# Insensitive Effects of Inflammatory Cytokines on the Reference Genes of Synovial Fluid Resident-Mesenchymal Stem Cells Derived from Rheumatoid Arthritis Patients

**DOI:** 10.3390/ijms242015159

**Published:** 2023-10-13

**Authors:** Eun-Yeong Bok, Saet-Byul Kim, Gitika Thakur, Yong-Ho Choe, Seong-Ju Oh, Sun-Chul Hwang, Sun-A. Ock, Gyu-Jin Rho, Sang-Il Lee, Won-Jae Lee, Sung-Lim Lee

**Affiliations:** 1College of Veterinary Medicine, Gyeongsang National University, Jinju 52828, Republic of Korea; eybok@gnu.ac.kr (E.-Y.B.); sbkim4@gnu.ac.kr (S.-B.K.); gitika18oct@gnu.ac.kr (G.T.); yhchoego@gmail.com (Y.-H.C.); osj414@gnu.ac.kr (S.-J.O.); jinrho@gnu.ac.kr (G.-J.R.); 2Department of Orthopaedic Surgery, Gyeongsang National University, School of Medicine and Hospital, Jinju 52727, Republic of Korea; hscspine@gnu.ac.kr; 3Animal Biotechnology Division, National Institute of Animal Science (NIAS), Rural Development Administration (RDA), 1500, Kongjwipatjwi-ro, Isero-myeon, Wanju-gun 565851, Republic of Korea; ocksa@korea.kr; 4Research Institute of Life Sciences, Gyeongsang National University, Jinju 52828, Republic of Korea; 5Department of Internal Medicine and Institute of Health Sciences, Gyeongsang National University School of Medicine and Hospital, Jinju 52727, Republic of Korea; goldgu@gnu.ac.kr; 6College of Veterinary Medicine, Kyungpook National University, Daegu 41566, Republic of Korea

**Keywords:** patient-derived mesenchymal stem cell, reference gene, rheumatoid arthritis, inflammatory cytokines, quantitative polymerase chain reaction

## Abstract

Mesenchymal stem cells derived from rheumatoid arthritis patients (RA-MSCs) provide an understanding of a variety of cellular and immunological responses within the inflammatory milieu. Sustained exposure of MSCs to inflammatory cytokines is likely to exert an influence on genetic variations, including reference genes (RGs). The sensitive effect of cytokines on the reference genes of RA-SF-MSCs may be a variation factor affecting patient-derived MSCs as well as the accuracy and reliability of data. Here, we comparatively evaluated the stability levels of nine RG candidates, namely *GAPDH*, *ACTB*, *B2M*, *EEF1A1*, *TBP*, *RPLP0*, *PPIA*, *YWHAZ*, and *HPRT1*, to find the most stable ones. Alteration of the RG expression was evaluated in MSCs derived from the SF of healthy donors (H-SF-MSCs) and in RA-SF-MSCs using the geNorm and NormFinder software programs. The results showed that *TBP*, *PPIA*, and *YWHAZ* were the most stable RGs for the normalization of H-SF-MSCs and RA-SF-MSCs using RT-qPCR, whereas ACTB, the most commonly used RG, was less stable and performed poorly. Additionally, the sensitivity of RG expression upon exposure to proinflammatory cytokines (TNF-α and IL-1β) was evaluated. RG stability was sensitive in the H-SF-MSCs exposed to TNF-α and IL-1β but insensitive in the RA-SF-MSCs. Furthermore, the normalization of *IDO* expression using *ACTB* falsely diminished the magnitude of biological significance, which was further confirmed with a functional analysis and an IDO activity assay. In conclusion, the results suggest that *TBP*, *PPIA*, and *YWHAZ* can be used in SF-MSCs, regardless of their exposure to inflammatory cytokines.

## 1. Introduction

Most research on patient-derived stem cells has been performed using embryonic stem cells (ESCs) and induced pluripotent stem cells (iPSCs) to screen and search for new drugs by modeling the cell to discover genetic disorders [1]. Although patient-derived ESCs and iPSCs can be used to identify the genetic causes of diseases, their ability to provide information about pathological cellular changes or clinical disease progression after disease onset is limited. However, patient-derived mesenchymal stem cells (MSCs) can provide an understanding of the various cellular and immunological responses to pathological or clinical disease-specific changes [2]. Moreover, patient-derived MSCs could serve as platforms for the delivery of biological agents into tumors in the carrier cells for inhibition [3] and could thus be used as a new class of medicines against autoimmune diseases such as rheumatoid arthritis (RA).

RA is characterized by chronic systemic synovitis, which causes cartilage and bone destruction via an interplay between infiltrating immune cells and resident stromal cells [4]. So far, the immunomodulation properties of MSCs have been successfully elucidated in RA animal models [5,6,7,8]. Therefore, the genetic profiling of MSCs is important for gaining an understanding of their role in the pathological regulation of RA. However, most studies have been performed using MSCs derived from healthy donors without any RA pathological events, and research involving MSCs derived from RA patients (RA-MSCs) is, to our knowledge, limited. MSCs are sources of progenitors for cell replacement and can activate or resuscitate other local cells, such as tissue-resident stem cells, endothelial cells, and fibroblasts, to facilitate tissue regeneration through paracrine stimulation [9,10,11,12]. The movement of MSCs to lesion sites caused by injury or disease is described as endogenous stem cell homing and migration, and the stimulation factors involved in tissue repair are known as natural self-healing responses. Various activation factors, proinflammatory cytokines or reactive oxygen species (ROS), and apoptotic factors can affect the functional activity of tissue-resident stem cells in local cells through migration and stimulation, and TNF-α and IL-1β are the main proinflammatory cytokines involved in the progression of RA [4,13]. Finding reliable and stable reference genes (RGs) for patient-derived MSCs, regardless of their exposure to proinflammatory cytokines, is important for understanding the pathogenesis and therapeutic applications. Therefore, analyzing the molecular characteristics of MSCs derived from the synovial fluid (SF) of inflammatory sites in RA patients (RA-SF-MSCs) is important to further understanding the genetic features and pathogenesis of RA, as these cells have been directly exposed to disease-specific conditions [2,8,14,15].

Real-time quantitative PCR (RT-qPCR) is the most basic technique used in MSC research and the most sensitive and reproducible analysis method for the evaluation of gene expression in MSCs. To gain accurate results, it is necessary to co-amplify an appropriate housekeeping gene as an endogenous reference during the transcriptional process, considering the diversity of individual samples and the enzymatic efficiency or impairment of techniques such as RNA extraction or cDNA synthesis. RGs should be expressed consistently under experimental conditions for the normalization of proteins and the mRNA expression analysis of target genes. RGs are expressed in the cells of all organisms and perform basic cellular functions like cell metabolism, proliferation, and the cell cycle, which are closely related to the properties of stem cells. Moreover, these genes are regulated in certain cell types and developmental stages and upon exposure to inflammatory environments [2,16,17]. This phenomenon can be attributed to the fact that RGs are involved in processes other than those related to their intrinsic functions [18].

In the present study, we investigated reliable and stable RGs for MSCs derived from patients’ lesion sites despite their exposure to proinflammatory cytokines, as stimulation or migration factors are important to various pathological and clinical application studies. We compared the stability levels of nine of the most well-known RGs, namely *GAPDH*, *ACTB*, *B2M*, *EEF1A1*, *TBP*, *RPLP0*, *PPIA*, *YWHAZ*, and *HPRT1*, in SF-MSCs derived from healthy donors and RA patients to select the most stable RGs in both MSC types. Furthermore, we evaluated the changes in RG expression levels upon the exposure to proinflammatory cytokines of SF-MSCs, under the assumption that the memory of an inflammatory stimulus may affect the stability of RGs. These stable RGs can be used to obtain reliable and accurate expression profiles of various genetic factors for understanding the pathological effects on MSCs in in vitro/in vivo experiments.

## 2. Results

### 2.1. Characterization of H-SF-MSCs and RA-SF-MSCs

The SF-MSCs isolated from healthy controls (H-SF-MSCs, *n* = 4) and RA patients (RA-SF-MSCs, *n* = 10) showed similar fibroblastic morphologies with plastic attachment ability (Figure 1A). Both types of SF-MSCs were successfully differentiated into mesenchymal lineages of adipocytes, osteoblasts, and chondrocytes under specific induction conditions; this was further confirmed with Oil red O, Alizarin red, and Alcian blue staining, respectively (Figure 1B). The MSC-specific markers (CD44, CD90, and CD105) were positively expressed, whereas the hematopoietic stem cell markers (CD34 and CD45) and MHC class II were negatively expressed in both types of SF-MSCs. There was no significant difference between the groups (Figure 1C).

We selected nine of the generally used housekeeping genes to find the most stably expressed RG in both types of SF-MSCs. These genes are functionally independent and not co-regulated, which is an essential prerequisite for geNorm analysis. Standard curves were produced to verify primer adequacy for RT-qPCR, which resulted in a correlation coefficient (R2) of 0.993–0.998 and PCR efficiencies (E) of 0.95–1.04 (Table 1). All RG candidates were amplified using a melting curve analysis with a single product, without nonspecific amplification (Figure 2A). In addition, all amplicons were visualized in a single band of the expected size, without any primer dimers, using gel electrophoresis (Figure 2B). 

### 2.2. CT Values of the RGs in the SF-MSCs upon Inflammatory Cytokine Induction

RT-qPCR was conducted to evaluate changes in the mRNA levels of the nine housekeeping genes under inflammatory cytokine treatment (TNF-α 50 ng/mL, IL-1β 50 ng/mL) for 48 h. All genes showed moderate cycle threshold (CT) values (all under 25 cycles) in both SF-MSCs (Figure 2C). The CT values of *GAPDH*, *ACTB*, *B2M*, and *HPRT1* in the H-SF-MSCs did not differ before and after treatment. However, the values of the other five genes (*EEF1A1*, *TBP*, *RPLP0*, *PPIA*, and *YWHAZ*) changed significantly (*p* = 0.02, *p* = 0.0037, *p* = 0.0033, *p* = 0.023, and *p* = 0.0389, respectively). In the RA-SF-MSCs, the CT values of *B2M* and *PPIA* changed significantly (*p* = 0.0019 and *p* = 0.0132, respectively), whereas the values of the other seven genes (*GAPDH*, *ACTB*, *EEF1A1*, *TBP*, *RPLP0*, *YWHAZ*, and *HPRT1*) did not. The standard deviations of the CT values across all samples showed that *PPIA*, *TBP*, and *YWHAZ* were the most stable genes, whereas *ACTB, B2M*, and *EEF1A1* were the most variable genes (Figure 2D). 

### 2.3. Analysis of the Most Stable Reference Gene Using geNorm

The CT values of the H-SF-MSCs and RA-SF-MSCs with or without inflammatory cytokine treatment were analyzed using geNorm. The MSCs were divided into three groups—H-SF-MSC, RA-SF-MSC, and H&RA-SF-MSC—and analyzed to obtain a stable ranking of the nine RGs from low M values to high M values. All M values were below 1.5, indicating that all of the genes used in this study were stable. The program results indicated that PPIA, YWHAZ, and TBP were the three most stable RGs in all SF-MSC groups, but ACTB, the most commonly used RG, was found to be unstable or less stable in all groups (Figure 3A). Additionally, the geNorm software also recommended the optimal number of RGs (NF_opt_) to be used for accurate normalization of the GOI by presenting pairwise variations (V). The NF_opt_ values for the H-SF-MSC, RA-SF-MSC, and H&RA-SF-MSC groups were V4/5, V8/9, and V8/9, respectively (Figure 3B). Because it is not realistic to use eight RGs for the normalization of the GOI in every experiment, the correlation between the three most stable RGs (NF3) and NF_opt_ was analyzed; all groups showed a high correlation (Pearson’s r > 0.95) between NF3 and NF_opt_ (Figure 3C). 

### 2.4. Analysis of the Most Stable Reference Gene Using NormFinder

The NormFinder results showed that *PPIA*, *YWHAZ*, and *TBP* were the most stable RGs in the H&RA-SF-MSCs, which is consistent with the geNorm results. However, in the H-SF-MSCs and RA-SF-MSCs, *ACTB*, which was among the top three in the geNorm analysis results, ranked fourth in the NormFinder analysis results. The geNorm and NormFinder rankings of the stable RGs differed slightly, which may be due to the NormFinder algorithms considering intergroup variations (Figure 4). Furthermore, the NormFinder program recommended the best combination of two genes from the nine RGs, specifically *YWHAZ* and *PPIA* for the H-SF-MSC and H&RA-SF-MSC groups and *YWHAZ* and *HPRT1* for the RA-SF-MSC group. 

### 2.5. The Impact of Using an Unstable Reference Gene in the Normalization of the GOI

To demonstrate the impact and importance of using the most stable RG for data normalization, we analyzed the relative expression of IDO, whose increased expression levels characterize the suppression effect on immune cell proliferation in SF-MSCs. The expression levels of IDO were assessed via RT-qPCR and normalized using the most and least stable RGs, as identified with geNorm and NormFinder. After TNF-α and IL-1β exposure, the expression levels of IDO in the T-H-SF-MSCs and T-RA-SF-MSCs significantly increased when *YWHAZ* (*p* < 0.0001 and *p* < 0.0001), *PPIA* (*p* < 0.0002 and *p* < 0.0001), and *TBP* (*p* < 0.0001 and *p* < 0.0001) were used for normalization (Figure 5A). This significant increase in expression at the mRNA level was corroborated using a functional assay; the production of L-kynurenine by IDO in the T-H-SF-MSCs (mean ± SD: 12.611 ± 0.561 μM) and T-RA-SF-MSCs (mean ± SD: 5.507 ± 1.360 μM) significantly increased compared with the control samples of H-SF-MSCs (mean ± SD: 0.119 ± 0.561 μM) and RA-SF-MSCs (mean ± SD: 0.047 ± 1.2646 μM) (Figure 5B). However, the use of ACTB, the least stable RG, showed no significant difference between the IDO expression levels of the RA-SF-MSCs and T-RA-SF-MSCs. The data demonstrated the importance of normalization using a stably expressed RG under specific experimental conditions to mimic the disease environment. Consequently, normalization using an inconsistent RG resulted in an underestimation of the magnitude of changes in the immunomodulation ability of the SF-MSCs, potentially affecting the interpretation of the biological significance of this study.

## 3. Discussion

The synovial cavity, which is filled with synovial fluid, is the site where autoimmune-mediated inflammation takes place in RA patients. Therefore, studying MSCs isolated from the synovial fluid of RA patients could provide an understanding of various cellular and immunological responses to pathological or clinical disease-specific changes [2]. Moreover, synovial fluid can be a useful cell source for autologous stem cell therapy because it can be easily obtained via syringe aspiration during diagnosis or treatment, and the population of MSCs is higher in injured joints than in healthy joints [19]. SF-MSCs are known to play an important role in the maintenance of homeostasis in the joint and to have immunosuppression capacity [19,20,21]. However, the chronic inflammation environment of the synovial fluid could affect SF-MSC characteristics [22]. For instance, the proliferation and chondrogenic differentiation ability of synovial MSCs decreases with an increase in the inflammation score of RA [21]. It has also been reported that when RA-SF-MSCs were classified based on disease duration and degree of joint destruction, the immunomodulatory capacity of early-standing patient-derived SF-MSCs reduced to a great extent after exposure to an RA inflammation stimulus compared with long-standing patient-derived SF-MSCs [2]. TNF-α and IL-1β are the main inflammatory cytokines in the progression of RA [4,13]; therefore, 50 ng/mL TNF-α and 50 ng/mL IL-1β are used to treat SF-MSCs and induce RA inflammation. 

RT-qPCR has been extensively utilized to elucidate the properties of MSCs and validate their therapeutic usefulness. The validation of stable RGs under experimental conditions is necessary to obtain reliable RT-qPCR results. Thus, in the present study, we established and investigated SF-MSCs and their CT values using *GAPDH*, *ACTB*, *B2M*, *EEF1A1*, *TBP*, *RPLP0*, *PPIA*, *YWHAZ*, and *HPRT1*, which are widely utilized RGs. A prerequisite for reliable gene expression studies on SF-MSCs is to find stably expressed RGs under critical experimental conditions, such as exposure to inflammatory cytokines. Therefore, to clarify how an RA-like environment affects the RGs of SF-MSCs, we analyzed the stability of RGs in SF-MSCs isolated from ten RA patients and compared them with SF-MSCs derived from four healthy individuals under general and RA inflammation culture conditions. To mimic the RA synovial environment, SF-MSCs were treated with TNF-α and IL-1β at 50 ng/mL each, determined by analyzing the synovial fluid of RA patients.

Previous reports have shown that RG expression levels may be affected by the experiment setup, including the culture conditions, growth factor treatment, and differentiation [16,23,24,25]. The present study demonstrated that the CT values and the standard deviation (SD) of nine RG candidates significantly changed after proinflammatory cytokine treatment in both H-SF-MSCs and RA-SF-MSCs. The two types of cells showed different gene expression tendencies. The H-SF-MSCs, which had never been exposed to inflammatory cytokines, displayed decreasing CT values after proinflammatory cytokine treatment. However, the RA-SF-MSCs, which had been continuously exposed to an in vivo inflammatory environment, showed an increase in gene expression level. Thus, the H-SF-MSCs and RA-SF-MSCs showed contrasting RG expression patterns (Figure 2C), indicating that it is necessary to find a stable RG by considering both types of cells after proinflammatory cytokine treatment. 

We evaluated the stability of the CT values by applying two software algorithms, namely geNorm and NormFinder, based on the most common combination for finding stably expressed RGs [26]. *PPIA*, *YWHAZ*, and *TBP* were determined to be the three most stable RGs for the normalization of gene expression in both types of SF-MSCs (H&RA-SF-MSCs) (Figure 3A and Figure 4). There was a subtle difference in these three RGs’ rankings in the H-SF-MSC and RA-SF-MSC groups, but the H&RA-SF-MSC group showed concordance between geNorm and NormFinder. Although several reports have validated stable RGs using other tissues from arthritis patients, this study is the first to find stable RGs in the SF-derived cells of RA patients. Schildberg et al., found new RGs suitable for the normalization of osteoarthritis (OA) patient-derived BM-MSCs: importin 8 (*IPO8*), cancer susceptibility candidate 3 (*CASC3*), and *TBP* [27]. In 2008, *TBP* was found to be the most stable gene in the cartilage tissues of osteoarthritis patients [28]. Another researcher found *PPIA* to be the stable RG in the bone MSCs of patients with avascular necrosis of the femoral head [29]. Watanabe et al., recommended *HPRT1*, *GAPDH*, and *YWHAZ* RGs for gene analysis in the synovium of OA patients [30]. 

Several studies on RG validation have been conducted using human MSCs and different experimental strategies. Tratwal et al., identified the RG combination of *TBP* and *YWHAZ* to be the most stable when using BM-MSCs and AD-MSCs regardless of the treatment approach, such as using the vascular endothelial growth factor (VEGF) to enhance the regeneration capability of MSCs [16]. *YWHAZ* and *PPIA* expression was stable in long-term cultured BM-MSCs and AD-MSCs [19]. Another researcher reported that *YWHAZ* was a stable RG in long-term cultured UC-MSCs [23]. 

*ACTB* and *B2M* are commonly used RGs that are inappropriate to use under various experimental conditions. In the present study, *ACTB* was the least stably expressed RG (Figure 3A and Figure 4). Similar to our findings using patient-derived MSCs, Schidberg et al., showed that *ACTB* and *B2M* were the most unstable RGs in BM-MSCs derived from OA patients [27]. An analysis of RGs in osteogenic differentiated BM-MSCs showed that *ACTB* displayed the greatest fold change in the GOI and was associated with morphological changes in the differentiated cells [24]. Further, fluctuations in *B2M* expression were found in the synovium of OA patients [30]. The instability of *ACTB* and *B2M* has been identified in various experimental conditions, including long-term cultured BM-MSCs and AD-MSCs, proinflammatory stimulators, LPS, and treated macrophages [19,31]. 

The recommended RG for RA animal models varies depending on the RA induction methods used and the target tissues. In a past study, *HPRT1, B2M*, and *RPL13A* were the stable genes in the arthritic joints of the KbxN serum transfer arthritis model, and the mRNA expression level of *GAPDH* significantly decreased with an increase in joint inflammation [32]. As a result of geNorm and NormFinder analysis, the expression levels of the housekeeping genes in various organs of a rat model induced by an adjuvant and those of the genes in the peripheral and neuronal tissues increased. Among them, *HPRT1* was the safest gene in all tissues except for the ankle joint. Due to excessive infiltration of inflammatory cells in the ankle joint, a stable RG could be found after normalization of the housekeeping gene with CD3, a representative T lymphocyte factor. *PPIB* is an RG that is constantly expressed, regardless of the presence or absence of arthritis [33]. A geNorm analysis of the spleen and inguinal lymph nodes of pristane-induced arthritis (PIA) rat models revealed that *18S*, *PPIA*, and *GUSB* were the three most stable genes in the spleen, whereas *18S*, *ACTB*, and *GAPDH* were the most stable genes in the inguinal lymph nodes [34]. 

The impact of normalizing expression data using an inconsistently expressed RG has been demonstrated in several publications [31,35]. A poorly selected RG in the normalization of a GOI using RT-qPCR can dramatically alter the results to a large extent and even lead to conflicting results. Usually, cytokines such as IFN-γ, TNF-α, and IL-1β are critical functional factors that trigger MSCs to induce the inhibition of cell proliferation and activation of immune cells via the secretion of anti-inflammatory molecules such as IDO [10,36,37]. However, we have previously found that SF-MSCs exposed to hypoxia and an inflammatory environment induce cellular senescence, which subsequently reduces their immunomodulatory ability. With the reduced immunomodulatory ability of SF-MSCs due to a high concentration of inflammatory cytokines, the IDO secretion ability of RA-SF-MSCs, which were exposed to an in vivo RA inflammatory environment for a long time, was lower than that of H-SF-MSCs. An IDO activity assay showed that, after treatment with inflammatory cytokines, the production of L-kynurenine, which is one of the metabolites in the IDO pathway, increased and was higher in H-SF-MSCs than in RA-SF-MSCs (fold change mean ± SD: 105.933 ± 4.244 μM and 5.507 ± 6.935 μM, respectively; Figure 5B). The IDO expression level significantly increased in the inflammatory cytokine-treated H-SF-MSCs and RA-SF-MSCs, compared with the nontreated groups, when normalized with *YWHAZ*, the most stable RG (*p* < 0.0001 and *p* = 0.0002), *PPIA* (*p* < 0.0001 and *p* < 0.0001), and *TBP* (*p* < 0.0001 and *p* < 0.0001). However, no significant difference in *IDO* expression was noted in the T-RA-SF-MSCs when *ACTB* was used for normalization (*p* = 0.0792, Figure 5A). This may be due to the large variation in *ACTB* expression between samples (SD = 2.143 in T-SF-MSCs) compared with other RGs (*YWHAZ* and *PPIA*, SD = 1.198 and 1.4, respectively). These results are consistent with the analysis shown in Figure 2D, as *ACTB* showed the largest standard variation across all samples. The present study showed that the impact of normalization using unstable RGs deceitfully moderated the magnitude of the biological significance. Therefore, it is necessary to identify the most stable RGs in each experimental design to obtain accurate gene analysis data. In conclusion, the present study demonstrated that three RGs are appropriate for normalizing the expression of the GOI, and the traditional RG *ACTB* is less stable under the current experimental conditions.

## 4. Materials and Methods

### 4.1. Chemicals and Ethics Used in the Experiments

All chemicals and media were purchased from Thermo Scientific (Waltham, MA, USA) or Qiagen (Hilden, Germany) unless otherwise specified. The collection of SF specimens was authorized under GNUH 2012-05-009 after obtaining informed consent from the patients.

### 4.2. Characterization of SF-MSCs from Healthy and RA Patients

SF was collected from four female donors with no signs of inflammatory joint disease (healthy donors; H-SF-MSCs) and ten female RA patients (RA-SF-MSCs) who were followed at Gyeongsang National University Hospital. The criteria for patient eligibility were disease activity score 28 (DAS28) ≥ 4.0 and the exclusion criteria were age ≤ 18 and other infectious diseases. The clinical histories of the RA patients are presented in Table 2. For cell isolation, SF was filtered through a 40 μm nylon cell strainer (BD Falcon, NJ, USA), and cell pellets were isolated with centrifugation at 400× *g* for 10 min. The cells were seeded onto 35 mm dishes (Nunc, Roskilde, Denmark) and allowed to adhere for two days in advanced Dulbecco’s modified Eagle medium (ADMEM) supplemented with 10% fetal bovine serum (FBS), 1% GlutaMaxTM, and 1% penicillin and streptomycin (10,000 IU and 10,000 μg/mL, respectively). The adherent cells were cultured in a culture medium at 36.5 °C using a humidified incubator with 5% CO_2_. Cells other than clusters were removed with a cell scraper following cell cluster formation, and the expanded cells were harvested through three passages.

Flow cytometry was performed to analyze the expression of MSC-specific surface markers (BD FACS LSRFortessa™, Becton Dickinson, Franklin Lakes, NJ, USA). SF-MSCs were harvested using 0.25% trypsin-EDTA and washed twice with DPBS. For CD44, CD90, CD34, and CD45 expression, the cells were incubated with anti-human FITC-conjugated antibodies; for CD105 and MHC class II expression, cells were incubated with anti-human APC-conjugated antibodies for 1 h at 4 °C. All antibodies were purchased from BD Pharmingen™ and diluted in 1% bovine serum albumin at a concentration of 1:100.

Further, to analyze the differentiation ability of the mesenchymal lineage, H- and RA-SF-MSCs were differentiated into adipocytes, osteocytes, and chondrocytes under a specific induction medium for three weeks, following previous studies [2]. Adipogenic or osteogenic differentiation was induced using Dulbecco’s modified Eagle medium (DMEM) supplemented with 10% FBS, 100 μM indomethacin, 10 μM insulin, and 1 μM dexamethasone or 10% FBS, 200 μM ascorbic acid, 10 mM glycerol-2-phosphate, and 0.1 μM dexamethasone, respectively. For chondrogenic differentiation, 2 × 10^5^ SF-MSCs were centrifuged for 5 min at 450× *g* in StemPro^®^ chondrogenesis differentiation medium with a 10% chondrogenesis supplement. The differentiation was confirmed with cytochemistry staining, using 0.5% Oil red O for adipocytes, 0.5% Alizarin red for osteoblasts, and 1% Alcian blue with 0.1% nuclear fast red solution for chondrocytes. All the experiments were performed at passage three and used for further analysis.

### 4.3. Treatment of H-SF-MSCs and RA-SF-MSCs with Pro-Inflammatory Cytokines

A total of 2 × 10^6^ SF-MSCs were seeded onto a 100 mm culture dish and allowed to adhere for 3 h before treatment with 50 ng/mL TNF-α and 50 ng/mL IL-1β (JW CreGene, Seongnam, Republic of Korea). This concentration was confirmed by analyzing the synovial fluid of RA patients (Quantikine^®^ ELISA kits, R&D Systems, Minneapolis, MN, USA). The cells were incubated with inflammatory cytokines for 48 h at 36.5 °C in 5% CO_2_. 

### 4.4. Candidate Reference Genes and Primer Sequences

The primers of nine RGs (Table 2) were chosen due to their different biological functions and irrelevance. Metabolism-related genes: glyceraldehyde-3-phosphate dehydrogenase (*GAPDH*), beta-2-microglobulin (*B2M*); ribosomal genes: ribosomal protein large P0 (*RPLP0*); cellular structure and cytoskeleton: beta-actin (*ACTB*); transcription and translation: eukaryotic translation elongation factor 1 alpha 1 (*EEF1A1*), TATA box binding protein (*TBP*); cell signaling and regulation: peptidylprolyl isomerase A (*PPIA*), tyrosine 3-monooxygenase/tryptophan 5-monooxygenase activation protein, zeta (*YWHAZ*); nucleotide metabolism: hypoxanthine phosphoribosyltransferase 1 (*HPRT1*). *GAPDH*, *B2M*, *EEF1A1*, RPLP0, *PPIA*, and *YWHAZ* were referenced from a previous report [35,38]. *ACTB*, *TBP*, and *HPRT1* were designed using the PrimerQuest™ tool at an annealing temperature of 60 °C and analyzed using OligoAnalyzer 3.1, a software that confirms the absence of hairpins, homodimers, and heterodimers. To evaluate primer efficiency, RT-qPCR was performed with the 10-fold diluted cDNA of the SF-MSCs, following a previously established protocol [38]. Standard curve parameters were calculated using the RotorGene Q Series software 2.1.0 (Qiagen, Hilden, Germany). 

### 4.5. RNA Extraction, cDNA Synthesis, and RT-qPCR

Total RNA was extracted from the SF-MSCs before and after inflammatory cytokine treatment using an easy-spin™ total RNA extraction kit (Invitrogen, Carlsbad, CA, USA). All RNAs were quantified using an OPTIGEN NanoQ Lite spectrophotometer (KLAB Co., Daejeon, Republic of Korea) and within 1.8–2.2 of the A260/A280 ratio. cDNA synthesis was performed with 0.5 μg total RNA at 60 °C for 1 h using a HisenScript™ RH (-) RT PreMix Kit (Invitrogen, Carlsbad, CA, USA). RT-qPCR was conducted using a Rotor-Gene Q (Qiagen, Hilden, Germany) RT-qPCR machine with a 2× SYBR Green PCR master mix (Qiagen, Hilden, Germany), with 50 ng cDNA per reaction and 0.7 μM forward and reverse primers. The RT-qPCR program involved pre-denaturation (95 °C for 2 min), two steps of 45 PCR cycles (denaturation at 95 °C for 5 s and combined annealing and extension at 60 °C for 10 s), and a melting curve analysis (from 60 °C to 90 °C at 1 °C/s). The amplification curve, melting curve, and cycle threshold values (CT values) were analyzed using the Rotor-Gene Series software (Qiagen, Hilden, Germany). In addition, the specificities and sizes of all amplicons were examined with gel electrophoresis using 1% agarose gel with 0.1 mg/mL ethidium bromide. 

### 4.6. Analysis of Stable Reference Gene Expression 

The stable RGs in the H-SF-MSCs and RA-SF-MSCs were comparatively analyzed. These groups included the generally cultivated samples and inflammatory cytokine-exposed samples. The CT values obtained from each RG were assessed using geNorm (version 3.3) and NormFinder (version 0.956), as previously described [39]. The geNorm software calculates gene expression stability by measuring M values; genes with high M values are sequentially removed until the most stable pair of RGs with the lowest M values are left. In addition, it provides a normalization factor (NF) for each RG and suggests the optimal number of RGs (NF_opt_) through a comparison of pairwise variations (V_n/n+1_) from NF_n_ to NF_n+1_. In the present study, the correlations between NF_opt_ and NF for the three most stable RGs (NF_3_) were analyzed using Pearson’s correlation on MS Excel to avoid the redundant use of RGs. NormFinder showed the stability rankings and standard errors of the RGs’ transcriptional variations through an analysis of variance (ANOVA)-based model to determine the single most stable RG and the recommended combination of two RGs.

### 4.7. Application of Different RGs to Normalize the Gene of Interest (GOI) 

To verify the effects of RG stability on GOI normalization, we used the most stable RGs (*PPIA, YWHAZ*, and *TPB*) and the least stable RG (*ACTB*), as determined by the geNorm and NormFinder analyses in this study. The relative gene expression of indoleamine 2,3-dioxygenase (*IDO*), a marker of the immune suppression ability of the MSCs, was evaluated using RT-qPCR before and after exposure to the H- and RA-SF-MSCs’ exposure to inflammatory cytokines. 

### 4.8. IDO Activity Measurements

An IDO activity assay was conducted following a previous protocol [2]. For 2 days, 2.5 × 10^4^ H-SF-MSCs and RA-SF-MSCs with or without induction of inflammatory cytokines (TNF-α 50 ng/mL, IL-1β 50 ng/mL) were cultured; they were then supplemented with 100 μM L-tryptophan (Sigma, Ronkonkoma, NY, USA) for 4 h. Supernatant was harvested from each culture well, and 30% trichloroacetic acid was added (Sigma, USA). This solution was incubated for 30 min at 50 °C and was diluted 1:1 in Ehrlich reagent (Sigma, USA). The optical density was measured at 492 nm using a microplate reader (Molecular Devices, San Jose, CA, USA). L-kynurenine (Sigma, USA) made with a fresh culture medium was serially diluted as the standard.

### 4.9. Statistical Analysis

A one-way ANOVA, Student’s *t*-test (CD markers and CT value), and Tukey’s post hoc test were applied to assess the relative expression of the GOI. A significant difference was determined among the groups using PASW Statistics 18 (SPSS Inc., IBM, Armonk, NY, USA). Significant differences were considered at *p* < 0.05, and all data were represented as mean ± standard deviation (SD) values.

## 5. Conclusions

In conclusion, this study sheds light on how synovial fluid-derived mesenchymal stem cells (SF-MSCs) interact with one another in the context of rheumatoid arthritis (RA), and it highlights the critical function of stable reference genes (RGs) in ensuring accurate gene expression analysis. The study underlines the necessity of careful RG selection to reduce the effects of various cell types and inflammatory conditions while highlighting the promise of SF-MSCs for therapeutic uses in RA, particularly in autologous stem cell treatment. This study provides crucial insights for improving the accuracy and reliability of gene expression investigations in the field of MSC research and treatment development by outlining the stability of RGs under both general and RA-specific conditions. 

## Figures and Tables

**Figure 1 ijms-24-15159-f001:**
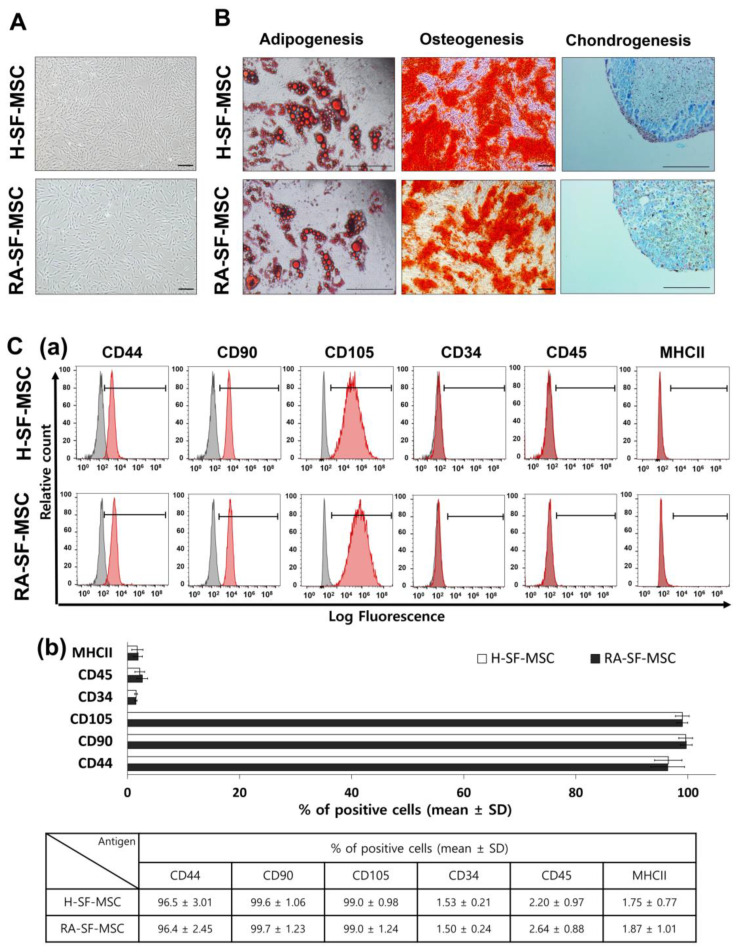
Characteristics and differentiation abilities of healthy donor or RA patient-derived SF-MSCs. (**A**) The fibroblastic morphology with plastic-adherent population; scale bar = 100 μm. (**B**) Lineage-specific differentiation of H-SF-MSCs and RA-SF-MSCs. Cytochemical staining of differentiated adipocytes (lipid droplets), osteoblasts (calcium deposits), and chondrocytes (proteoglycan synthesis) when grown in vitro; scale bar = 250 μm. (**C**(**a**)) Cell surface marker analysis of MSC-positive markers (CD44, CD90, and CD105) and MSC-negative markers (CD34, CD45, and MHC-II). Peaks in black color indicate control and red color indicates SF-MSCs. (**C**(**b**)) The graphs are presented as percentage mean ± SD. H-SF-MSC, healthy donors-derived SF-MSCs; RA-SF-MSC, RA patients-derived SF-MSCs.

**Figure 2 ijms-24-15159-f002:**
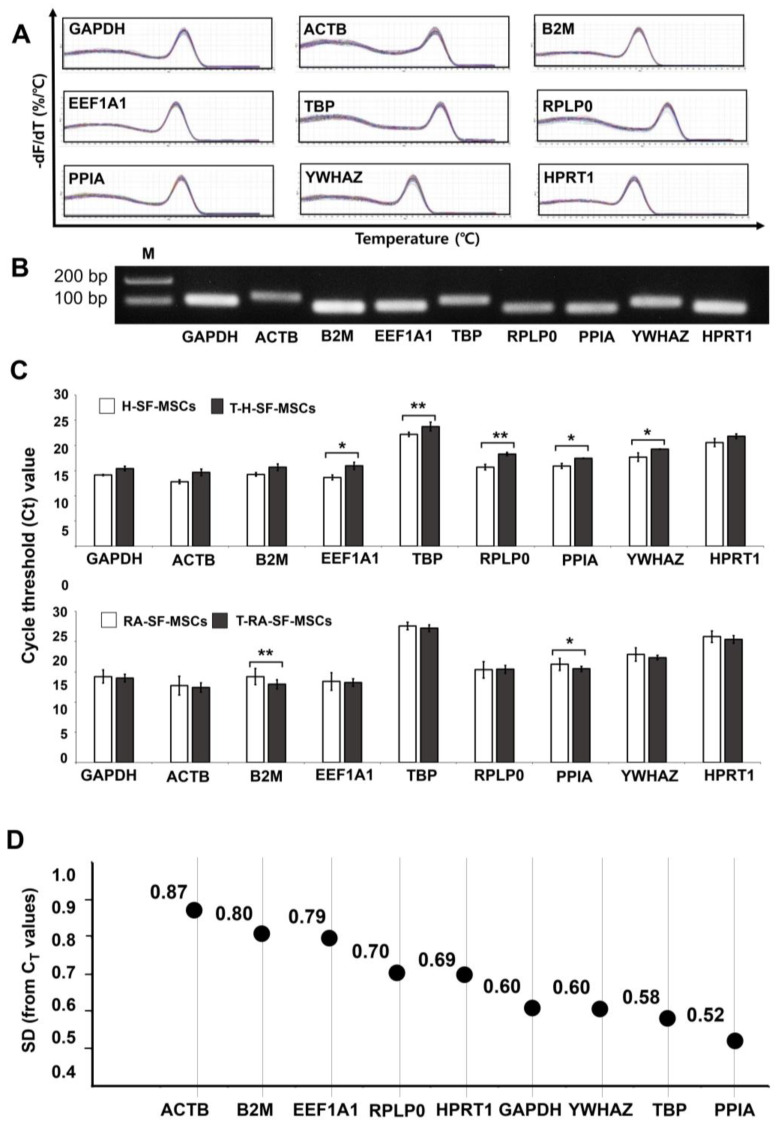
Examination of primer specificity, amplicon size, and CT values of selected RGs. (**A**) The specificity of nine candidates of RGs was analyzed using melting curves that were obtained by plotting the negative derivative of fluorescence over temperature (-dF/dT) versus temperature (from 60 °C to 90 °C at 1 °C/s). (**B**) The length of the amplified products of nine RGs was analyzed using agarose gel electrophoresis (**C**) The CT values of nine candidates of RGs in H-SF-MSCs and RA-SF-MSCs with or without inflammatory cytokines treatment. The data are presented mean ± SD; * *p* < 0.05; ** *p* < 0.01 vs. H-SF-MSCs or RA-SF-MSCs. (**D**) The standard deviation of CT values for each of the RGs across all samples. T-H-SF-MSC, healthy donors-derived SF-MSCs treated with 50 ng/mL TNF-α and 50 ng/mL IL-1β for 48 h; T-RA-SF-MSC, RA patients-derived SF-MSCs treated with 50 ng/mL TNF-α and 50 ng/mL IL-1β for 48 h.

**Figure 3 ijms-24-15159-f003:**
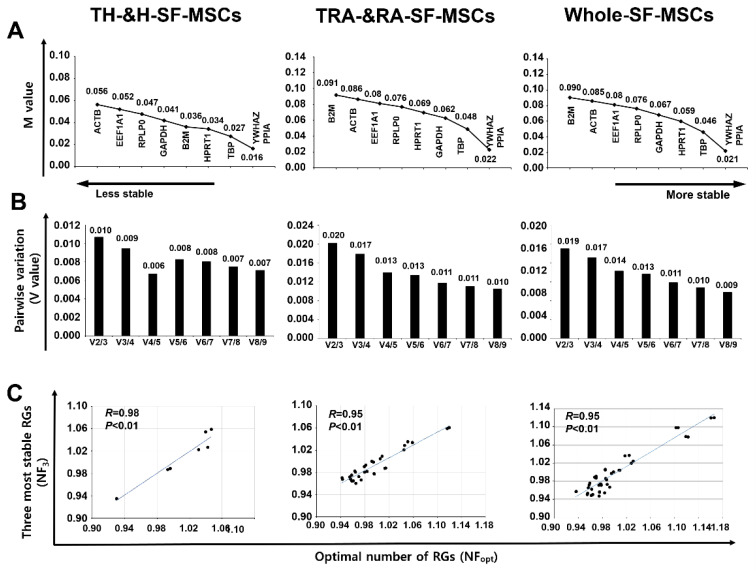
Analysis of the most stable RG using geNorm. (**A**) Evaluated M values of nine RGs; the x-axis from right to left indicates the stability ranking. (**B**) An optimal number of recommended RGs for normalization (NF_opt_); software calculated the normalization factors (NFs) from at least two genes where variable V defines the pair-wise variation between two sequential NFs. (**C**) Correlation between NF_opt_ and NF3. Pearson’s correlation was calculated using Excel software; NF3, the three most stable RGs for normalization. Whole-SF-MSC includes H-SF-MSCs, T-H-SF-MSCs, RA-SF-MSCs, and T-RA-SF-MSCs.

**Figure 4 ijms-24-15159-f004:**
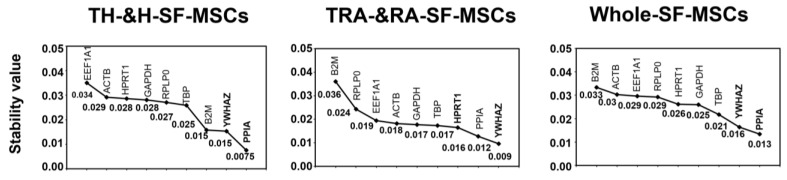
Analysis of the most stable RG using NormFinder. The stability values of nine RGs in SF-MSCs were calculated with NormFinder. Lower stability values indicate more stably expressed genes. The x-axis from right to left indicates the ranking of the genes according to expression stability.

**Figure 5 ijms-24-15159-f005:**
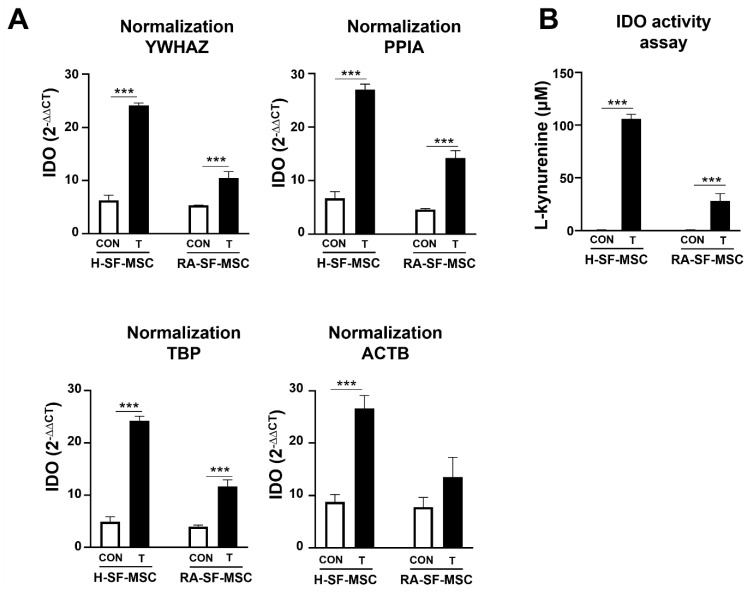
The impact of using an unstable RG in the normalization of GOI. (**A**) IDO gene expression level in SF-MSCs was normalized using the most stable RGs (*PPIA*, *YWHAZ*, and *TBP*) validated with geNorm and NormFinder in whole-SF-MSCs, and the traditional RG (ACTB) to demonstrate the effect of validated RGs on the normalization of a gene of interest (GOI). (**B**) The IDO activity assay; the production of L-kynurenine from L-tryptophan, which is metabolized by IDO. The data are presented as mean ± SD; *** *p* < 0.001 vs. H-SF-MSCs or RA-SF-MSCs; IDO, indoleamine 2,3-dioxygenase.

**Table 1 ijms-24-15159-t001:** Candidate reference genes and their information.

Primer Information				Standard Curve Parameters				
Gene Name (Symbol)		Sequence	Base Pair	Accession	R^2^	M	B	E
Actin beta (*ACTB*)	F:	TCCACGAAAC TACCTTCAACTC	131	NM_001101.5	0.996	−3.514	34.524	0.95
	R:	GATCTCCTTCT CATCCTGTCG						
Beta-2-microglobulin (*B2M*)	F:	CAGCTACTCCA AAGATTCAGG	116	AF_072097.1	0.993	−3.356	39.045	1.04
	R:	GGATGAAACCC AGACACATAGC						
Eukaryotic translation elongation factor 1 alpha 1 (*EEF1A1*)	F:	ACTATCATGAT GCCCCAGGAC	121	NM_001402.5	0.997	−3.356	32.288	0.96
	R:	ACACCAGCAG CAACAATCAG						
Glyceraldehyde-3-phosphate dehydrogenase (*GAPDH*)	F:	GTCAACGGATTTGGTCGTATTGG	108	NM_002046.7	0.995	−3.499	33.652	1.03
	R:	CATGTAGTTGAG GTCAATGAAGGG						
Hypoxanthine phosphoribosyltransferase 1 (*HPRT1*)	F:	CTGGCGTCGTGA TTAGTGAT	90	M_31642.1	0.996	−3.485	32.587	1.01
	R:	ACACCCTTTCCA AATCCTCA						
Peptidylprolyl isomerase A (*PPIA*)	F:	TGCTGGACCCA ACACAAATG	89	NM_001002.3	0.995	−3.468	32.145	0.97
	R:	AACACCACATG CTTGCCATC						
Ribosomal protein, large, P0 (*RPLP0*)	F:	TGGGCAAGAAC ACCATGATG	98	NM_001402.5	0.997	−3.358	38.541	1.03
	R:	TTTGTGGGACAGCATGGATG						
Tyrosine 3-monooxygenase/tryptophan 5-monooxygenase activation protein, zeta (*YWHAZ*)	F:	CGAAGCTGAAG CAGGAGAAG	111	NM_021130.3	0.995	−3.477	35.568	1.01
	R:	TTTGTGGGACAG CATGGATG						

**Table 2 ijms-24-15159-t002:** Demographic and clinical data of donors.

Characteristics	H-SF-MSCs (*n* = 4)	RA-SF-MSCs (*n* = 10)	*p* Value *
Age (years)	30.5 (3.69)	58.8 (2.5)	0.0002
Women	4 (100%)	10 (100%)	-
Positive for rheumatoid factor	-	8 (80%)	-
Positive for anti-CCP antibody	-	8 (80%)	-
DAS28-ESR	-	4.75 (0.3)	-
Disease duration (years)	-	7.9 (2.9)	-

CCP, cyclic citrullinated peptide; DAS28-ESR, disease activity score 28-erythrocyte sediment rate. Data are means (SE) or n (%), * *p* value indicates H-SF-MSC vs. RA-SF-MSC.

## Data Availability

Available on request if required.

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
