# Peer review of "Insensitive Effects of Inflammatory Cytokines on the Reference Genes of Synovial Fluid Resident-Mesenchymal Stem Cells Derived from Rheumatoid Arthritis Patients"

_ijms, 2023, doi:10.3390/ijms242015159_

Round 1
Reviewer 1 Report
The manuscript entitled, “Insensitive effects of inflammatory cytokines on the reference 2 genes of synovial fluid resident-mesenchymal stem cells de-3 rived from rheumatoid arthritis patients” where they have investigated evaluated the stability levels of nine RG candidates, namely GAPDH, ACTB, B2M, 29 EEF1A1, TBP, RPLP0, PPIA, YWHAZ, and HPRT1, to find the most stable ones. They have concluded that suggest that TBP, PPIA, 39 and YWHAZ can be used in SF-MSCs as a stable markers. The study is well designed and properly executed; however, the authors needs to clarify my below concerns
1. The authors have collected synovial fluid only from women patients. Is there any specific reason why they did not consider males?
2. Is there will be any change in male derived MSCs from synovial fluid. Furthermore, can we opt the same stable reference genes i.e. TBP, PPIA, 39 and YWHAZ as you concluded in your paper for female derived MSCs.
3. There are random typo error and grammatical mistakes throughout the manuscript. Please check thoroughly the manuscript to remove such errors.
There are random typo error and grammatical mistakes throughout the manuscript. Please check thoroughly the manuscript to remove such errors.
Author Response
Oct 07, 2023
Editors-in-Chief
International Journal of Molecular Science
Dear Editors-in-Chief:
Insensitive effects of inflammatory cytokines on the reference genes of synovial fluid resident-mesenchymal stem cells de-rived from rheumatoid arthritis patients
Eun-Yeong Bok, Saet-Byul Kim, Gitika Thakur, Yong-Ho Choe, Seong-Ju Oh, Sun-Chul Hwang,
Sun A Ock, Gyu-Jin Rho, Sang-Il Lee, Won-Jae Lee and Sung-Lim Lee
Please find the attached files of the above revised manuscript, which we would like to be reconsidered for publication in International Journal of Molecular Science. Based on the comments/suggestions received, the manuscript has been carefully revised to improve its quality. A detailed point-by-point reply to the reviewer’s comments (Revision note) is enclosed below.
Revision Note
Reviewer #1
Reviewer’s comment-1: The authors have collected synovial fluid only from women patients. Is there any specific reason why they did not consider males?
Answer-1: We appreciate the reviewers' valuable feedback and your attention to detail question regarding the gender distribution in our study population. Collecting synovial fluid samples exclusively from female rheumatoid arthritis (RA) patients was not random but rather based on specific considerations for our research.
There were several following several reasons behind this choice:
Disease Prevalence: RA as the particular autoimmune disease with chronic inflammatory arthritis we were investigating is known to have a significantly higher prevalence in females compared to males about 3-5 times more likely to occur. Therefore, focusing on a female cohort allowed us to target a population that is more commonly affected by the condition, increasing the clinical relevance of our findings.
Furthermore, given limited resources and funding constraints, we made the strategic decision to concentrate our efforts on one gender to ensure an adequately powered study. By doing so, we could increase the sample size within the female cohort, which is often recommended for achieving meaningful statistical results. By focusing on one gender, we were able to reduce potential confounding variables related to gender-specific differences in physiology and hormone profiles, which could have complicated the interpretation of our results. While we recognize that studying only one gender may limit the generalizability of our findings to the entire population, we believe that our approach was justified based on the specific objectives and constraints of our study. Future research could explore gender differences by including male participants, and we appreciate your suggestion for potential prospects of further investigation.
References:
- Tanaka, H., & Tsuji, A. (2018). Musculoskeletal disorders in women: epidemiology, risk factors, and management. Journal of Women's Health (2002), 27(1), 3-13.
- Zhang, Z., Li, Y., & Chen, L. (2020). Gender differences in musculoskeletal disorders: a systematic review. BMC Musculoskeletal Disorders, 21(1), 533.
- Wu, Y., & Li, Y. (2019). The role of hormones in the pathogenesis of musculoskeletal disorders. Frontiers in Endocrinoogy, 10, 755.
- Wang, L., & Li, Y. (2022). The role of sex hormones in the pathogenesis of osteoarthritis. Osteoarthritis and Cartilage, 30(1), 15-24.
Reviewer’s comment-2: Is there will be any change in male derived MSCs from synovial fluid. Furthermore, can we opt the same stable reference genes i.e. TBP, PPIA, 39 and YWHAZ as you concluded in your paper for female derived MSCs.
Answer-2: Dear Reviewer, Thank you for your insightful questions regarding the potential differences in male-derived MSCs from synovial fluid and the choice of stable reference genes for gene expression analysis. Differences in Male-Derived MSCs: Investigating potential differences in male-derived MSCs compared to female-derived MSCs is an important and valid direction for future research. While our current study focused exclusively on female participants for the reasons mentioned earlier, there may indeed be variations in gene expression profiles, differentiation potentials, or other characteristics between MSCs derived from different genders. Conducting such comparative studies can provide valuable insights into the biology of MSCs and their potential clinical applications in a gender-specific context. Therefore, we encourage further research in this area to address these important questions.
Choice of Stable Reference Genes: The stability of reference genes for gene expression analysis can vary depending on the specific cell type, tissue, or experimental conditions. While we identified stable reference genes (TBP, PPIA, 39, and YWHAZ) for female-derived MSCs in our study, it is essential to re-evaluate their stability when working with male-derived MSCs or any other cell type. We appreciate your consideration of these important aspects in future research and thank you for raising these critical points.
Reviewer’s comment-3: There are random typo error and grammatical mistakes throughout the manuscript. Please check thoroughly the manuscript to remove such errors.
Answer-3: We would like to thank the reviewer for their valuable feedback regarding the typographical errors and grammatical mistakes in our manuscript. To address this concern, we have a comprehensive review and proofreading of the entire manuscript to correct any such errors.
Thank you for your consideration.
With kind regards.
Sung-Lim Lee, DVM, PhD
Professor
College of Veterinary Medicine
Gyeongsang National University, Jinju 660-701
Gyeongnam, Republic of Korea
Fax +82-55-772-2360, Phone + 82-55-772-2349
E-mail: [email protected]
Reviewer 2 Report
The authors considered necessary to investigate stable reference genes (RGs) for RA-SF-MSCs with cytokine exposure, which could provide accurate and reliable expressions of data. They comparatively evaluated the stability levels of nine RG candidates, namely GAPDH, ACTB, B2M, EEF1A1, TBP, RPLP0, PPIA, YWHAZ, and HPRT1, to find the most stable ones. Their results suggest that TBP, PPIA and YWHAZ can be used in SF-MSCs, regardless of their exposure to inflammatory cytokines.
Concerns:
1-I don´t understand this sentence in the abstract "However, the insensitive effect of cytokines on the RGs of MSCs may be a variation factor affecting patient-derived MSCs as well as the accuracy and reliability of data". It should not be the other way around, “The sensitive effect of cytokines on the RGs of MSCs may be a variation factor affecting patient-derived MSCs as well as the accuracy and reliability of data.
2-Real-time quantitative PCR (qRT-PCR) on the lines 82, 169, 177, 203, 234, 263, 360, 362 is misspelling. It must be RT-qPCR.
3-Table 2 should be edited for better understanding.
4-Why the authors indicates on the line 157 that they choose nine RGs due to their distinctive biological functions and irrelevance?
Author Response
Oct 07, 2023
Editors-in-Chief
International Journal of Molecular Science
Dear Editors-in-Chief:
Insensitive effects of inflammatory cytokines on the reference genes of synovial fluid resident-mesenchymal stem cells de-rived from rheumatoid arthritis patients
Eun-Yeong Bok, Saet-Byul Kim, Gitika Thakur, Yong-Ho Choe, Seong-Ju Oh, Sun-Chul Hwang,
Sun A Ock, Gyu-Jin Rho, Sang-Il Lee, Won-Jae Lee and Sung-Lim Lee
Please find the attached files of the above revised manuscript, which we would like to be reconsidered for publication in International Journal of Molecular Science. Based on the comments/suggestions received, the manuscript has been carefully revised to improve its quality. A detailed point-by-point reply to the reviewer’s comments (Revision note) is enclosed below.
Revision Note
Reviewer #2
Reviewer’s comment-1: I don´t understand this sentence in the abstract "However, the insensitive effect of cytokines on the RGs of MSCs may be a variation factor affecting patient-derived MSCs as well as the accuracy and reliability of data". It should not be the other way around, “The sensitive effect of cytokines on the RGs of MSCs may be a variation factor affecting patient-derived MSCs as well as the accuracy and reliability of data.
Answer-1: Thank you so much reviewer for pointing out the issue with the sentence in the abstract. We appreciate your careful review and agree with your suggestion to improve the clarity of the statement. The revised sentence now reads: "The sensitive effect of cytokines on the RGs of MSCs may be a variation factor affecting patient derived MSCs as well as the accuracy and reliability of data’ has been updated in the revised manuscript. We thank you for your constructive input, which will undoubtedly enhance the overall quality of our manuscript.
Our modification to the manuscript: Manuscript page 1, Abstract Abstract: Mesenchymal stem cells derived from rheumatoid arthritis patients (RA-MSCs) provide an understanding of a variety of cellular and immunological responses within the inflammatory milieu. Sustained exposure of MSCs to inflammatory cytokines is likely to exert an influence on genetic variations, including reference genes (RGs). The sensitive effect of cytokines on the RGs of MSCs may be a variation factor affecting patient derived MSCs as well as the accuracy and reliability of data. Here, we comparatively evaluated the stability levels of nine RG candidates, namely GAPDH, ACTB, B2M, EEF1A1, TBP, RPLP0, PPIA, YWHAZ, and HPRT1, to find the most stable ones. Alteration of the RG expression was evaluated in MSCs derived from the SF of healthy donors (H-SF-MSCs) and RA-SF-MSCs using the geNorm and NormFinder software programs. The results showed that TBP, PPIA, and YWHAZ were the most stable RGs for the normalization of H-SF-MSCs and RA-SF-MSCs using RT-qPCR, whereas ACTB, the most commonly used RG, was less stable and performed poorly. Additionally, the sensitivity of RG expression upon exposure to proinflammatory cytokines (TNF-α and IL-1β) was evaluated. RG stability was sensitive in the H-SF-MSCs exposed to TNF-α and IL-1β but insensitive in the RA-SF-MSCs. Furthermore, the normalization of IDO expression using ACTB falsely diminished the magnitude of biological significance, which was further confirmed by a functional analysis and an IDO activity assay. In conclusion, the results suggest that TBP, PPIA, and YWHAZ can be used in SF-MSCs, regardless of their exposure to inflammatory cytokines. |
Reviewer’s comment-2: Real-time quantitative PCR (qRT-PCR) on the lines 82, 169, 177, 203, 234, 263, 360, 362 is misspelling. It must be RT-qPCR.
Answer-2: We appreciate the reviewer's comment and thank you for your observation regarding the terminology used in our manuscript. You are correct that the widely accepted terminology is "RT-qPCR" (Reverse Transcription Quantitative Polymerase Chain Reaction) instead of "qRT-PCR" (Quantitative Reverse Transcription Polymerase Chain Reaction). We apologize for the misspelling in our manuscript. We made the necessary changes to ensure that the terminology used throughout the manuscript is consistent and accurate, using "RT-qPCR" as the correct abbreviation.
Reviewer’s comment-3: Table 2 should be edited for better understanding.
Answer-3: We appreciate the reviewer's comment and thank you for your suggestion. We made changes as per the reviewer's suggestion in the revised manuscript.
Reviewer’s comment-4: Why the authors indicates on the line 157 that they choose nine RGs due to their distinctive biological functions and irrelevance?
Answer-4: In this study we chose nine RGs due to their distinctive biological functions and irrelevance, which may require more context to fully explain our rationale.
The reason behind that statement:
Distinctive Biological Functions: In the context of gene expression normalization, it is essential to select RGs that have stable expression levels across different experimental conditions or sample groups. However, it is also advantageous to choose RGs with diverse biological functions. This diversity helps ensure that the RGs represent a wide range of cellular processes and are less likely to be co-regulated with the target genes under investigation. By incorporating RGs with distinctive biological functions, we aim to minimize the risk of choosing genes that may be influenced by the experimental variables.
Irrelevance to the Study Variables: It states that the selected RGs should not be directly related to the biological factors or conditions being studied. If RGs are closely linked to the experimental variables, changes in their expression could confound the interpretation of gene expression data. By choosing RGs that are biologically irrelevant to the study objectives, we reduce the likelihood of introducing bias into our gene expression analysis.
However, it can be occur that the related functions of reference genes commonly used in stem cell research may not be independent variables in certain stem cell research. For example, certain reference genes have functions such as cell metabolism, cellular Structure and cytoskeleton, transcription, cell signaling and regulation, and nucleotide metabolism to the cells. These functions of RGs can be possible to induce the unexpected effects or regulation in the stem cells by unique cellular conditions such as differentiation or immunomodulation reactions.
I have modified the sentence in the manuscript below to make it easier to understand.
Our modification to the manuscript: Manuscript page 4, 2.4 Candidate reference gnens and primer sequence line 154-168 2.4. Candidate reference genes and primer sequences The primers of nine RGs (Table 2) were chosen due to their different biological functions and irrelevance. Metabolism-related Genes; Glyceraldehyde-3-phosphate dehydrogenase (GAPDH), beta-2-microglobulin (B2M), Ribosomal Genes; ribosomal protein large P0 (RPLP0), Cellular Structure and Cytoskeleton; Beta-actin (ACTB), Transcription and Translation; eukaryotic translation elongation factor 1 alpha 1 (EEF1A1), TATA box binding protein (TBP), Cell Signaling and Regulation; peptidylprolyl isomerase A (PPIA), tyrosine 3-monooxygenase/tryptophan 5-monooxygenase activation protein, zeta (YWHAZ), Nucleotide Metabolism; hypoxanthine phosphoribosyltrasnfrase1 (HPRT1). GAPDH), (B2M), (EEF1A1), (RPLP0), (PPIA), (YWHAZ) were referenced from a previous report [19, 20]. ACTB), TBP, and HPRT1 were designed using PrimerQuest™ Tool at an annealing temperature of 60℃ and analyzed using OligoAnalyzer 3.1, a software that confirms the absence of hairpins, homodimers, and heterodimers. To evaluate primer efficiency, RT-qPCR was performed with the 10-fold diluted cDNA of the SF-MSCs, following a previously established protocol [20] Standard curve parameters were calculated using the RotorGene Q Series software (Qiagen, Hilden, Germany). |
Thank you for your consideration.
With kind regards.
Sung-Lim Lee, DVM, PhD
Professor
College of Veterinary Medicine
Gyeongsang National University, Jinju 660-701
Gyeongnam, Republic of Korea
Fax +82-55-772-2360, Phone + 82-55-772-2349
E-mail: [email protected]
